# Single nucleus RNA sequencing of the cerebral cortex from genetically diverse inbred mouse strains reveals differences in pericyte and endothelial cell composition

Jieun Park[1,2], Bonnie Taylor-Blake[1,3], James L. Krantz[1,3], Mark J. Zylka [1,2,3]*

1 UNC Neuroscience Center, The University of North Carolina at Chapel Hill, Chapel Hill, North Carolina, United States of America, 2 Carolina Institute for Developmental Disabilities, The University of North Carolina at Chapel Hill, Chapel Hill, North Carolina, United States of America, 3 Department of Cell Biology and Physiology, The University of North Carolina at Chapel Hill, Chapel Hill, North Carolina, United States of America

* zylka@med.unc.edu

## Abstract

Genetic background influences animal behavior and susceptibility to brain diseases. To evaluate how genetic background influences the relative number and/or types of cells in the brain, we performed single nucleus RNA sequencing (snRNAseq) on fourteen genetically distinct collaborative cross (CC) inbred mouse strains. These data comprise over 287,000 nuclei derived from 92 samples across the strains. We identified 12 principal cell types and 60 refined cell types in each of the strains. Pericytes and endothelial cells were the two principal cell types to show a statistically significant difference in cell proportion between strains. We validated these findings histologically by staining for pericyte (CD13) and endothelial cell (PECAM-1) markers. Consistent with our snRNAseq analyses, we histologically observed differences in pericyte and endothelial cell counts between strains. In addition, we found that the proportion of certain subsets of excitatory and inhibitory neurons varied in some strains. Overall, our study suggests that genetic background can influence brain cell type composition, with a notable influence on cells that make up the neurovascular unit.

## Introduction

Single-cell RNA sequencing (scRNAseq) technologies have been used to identify and characterize a diverse array of brain cell types in mice, revealing insights into cell type composition across brain regions, developmental trajectories, and the dynamic responses to stimuli or disease [1–7]. These studies made use of either inbred C57BL/6 mice or outbred CD-1 mice, raising the question as to whether these findings generalize to additional mouse genetic backgrounds.

**Data availability statement:** The raw and processed sequencing data generated in this study have been deposited in the Gene Expression Omnibus (GEO) database under accession number GSE263649. The metadata and scripts used for snRNAseq analysis are available in the GitHub page (https://github.com/jieunesther/CC_snRNAseq/).

**Funding:** J.P. is supported by grants from the National Institute of Child Health and Human Development (NICHD; T32HD040127). Microscopy was performed at the UNC Neuroscience Microscopy Core (RRID:SCR_019060), supported, in part, by funding from the NIH-NINDS (P30 NS045892) and the NIH-NICHD (P50 HD103573). M.J.Z. is supported by grants from the NINDS, NIMH, and NIEHS (R01NS109304, R01MH120229, R35ES028366). The funders had no role in study design, data collection and analysis, decision to publish, or preparation of the manuscript.

**Competing interests:** The authors declare no competing interests.

In humans, functional genomics data from 1,866 brain samples revealed that several brain disorders were associated with alterations in brain cell type proportions [8]. Different inbred mouse strains likewise show profound phenotypic differences in behavior and susceptibility to disease [9–13]. For instance, Gu et al identified CC strains with extreme seizure resistance and SUDEP (sudden unexpected death in epilepsy) susceptibility and genomic loci associated with seizure sensitivity using standardized seizure assays and quantitative trait loci (QTL) mapping [11]. Similarly, inbred strains such as C57BL/6NCrl and 129S2/SvPasCrl display marked differences in hippocampal gamma oscillation power and sharp wave-ripple events compared to C57BL/6J [14]. Additionally, anatomical variation in cortical area maps between C57BL/6J and DBA/2J highlights the influence of genetic background on brain structure [15]. While these phenotypic differences could, in part, relate to differences in brain cell composition, the extent to which strain background influences the types and proportions of cell types in the mouse brain has not been studied.

Collaborative Cross (CC) mice are a panel of recombinant inbred strains known for their genetic diversity [16]. These mice were derived from eight inbred founder strains, including strains that are commonly used in neuroscience research (C57BL/6J, 129S1/SvImJ, and A/J). Here, we sought to examine fourteen genetically distinct CC strains to gain insights into the spectrum of brain cell types and their variations across inbred mouse strains. Our study reveals that mouse genetic background impacts the proportions of pericytes and endothelial cells, components of the neurovascular unit, with potential implications for physiological processes, drug delivery through the blood brain barrier (BBB), and diseases that are associated with neurovascular biology.

## Results

### Classification of brain cell types in fourteen different genetically diverse collaborative cross mouse strains

To investigate the influence of genetic background on brain cell type diversity and composition, we performed single nucleus RNA sequencing (snRNAseq) from cerebral cortex tissue of fourteen collaborative cross (CC) inbred mouse strains (Fig 1A). We identified a comparable number of UMIs/nucleus and number of genes/nucleus from each strain after sequencing and filtering (see Methods; S1A, S1B Fig). To further quality control the data before clustering, we plotted each sample by the normalized counts of five sex chromosome genes (one X chromosome gene and four Y chromosome genes) to confirm that all our samples were from males (S1E Fig), the sex we selected for this study. One sample (CC084 #3) had a high number of normalized counts to *Xist*, a gene on the X chromosome that is expressed in females, as well as to genes on the Y chromosome (S1E, S1F Fig), which we confirmed by qPCR (S1G Fig). These data suggest that this animal, which we excluded from further analysis, was XXY. In humans, the presence of an extra X chromosome causes Klinefelter syndrome, a disorder that affect 0.1-0.2% of all males [17].

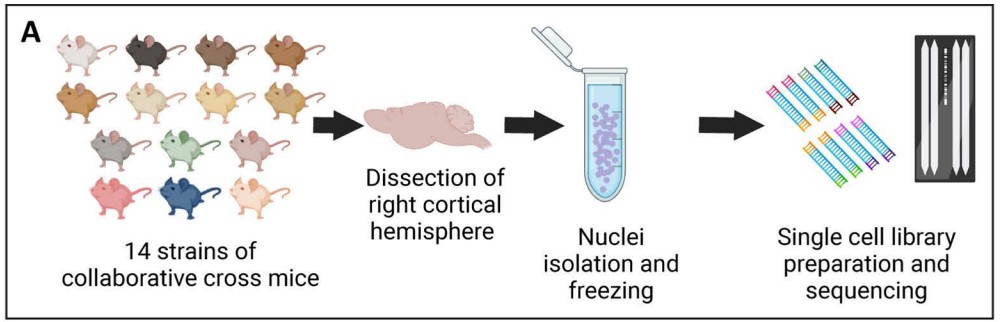

**Fig 1. Classification of 12 principal brain cell types in CC mouse strains using snRNAseq. (A)** Schematic showing sample preparation and snRNA sequencing process. **(B)** UMAP showing 12 principal cell types. **(C)** Dotplot showing major markers for each annotated principal cell type. **(D)** FeaturePlot of major cell type marker genes, showing the UMAP location of each cell type.

After unsupervised cell clustering, we detected 12 principal cell types (Fig 1B) and annotated these cell types using marker genes described in other mouse brain single cell RNAseq (scRNAseq) datasets (Fig 1C, 1D). The number of UMIs/nucleus and number of genes/nucleus from each principal cell type is shown in S1C, S1D Fig. Additionally, we annotated the clusters into 60 refined cell types (Fig 2A) that could be hierarchically organized (Fig 2C) by correlating our

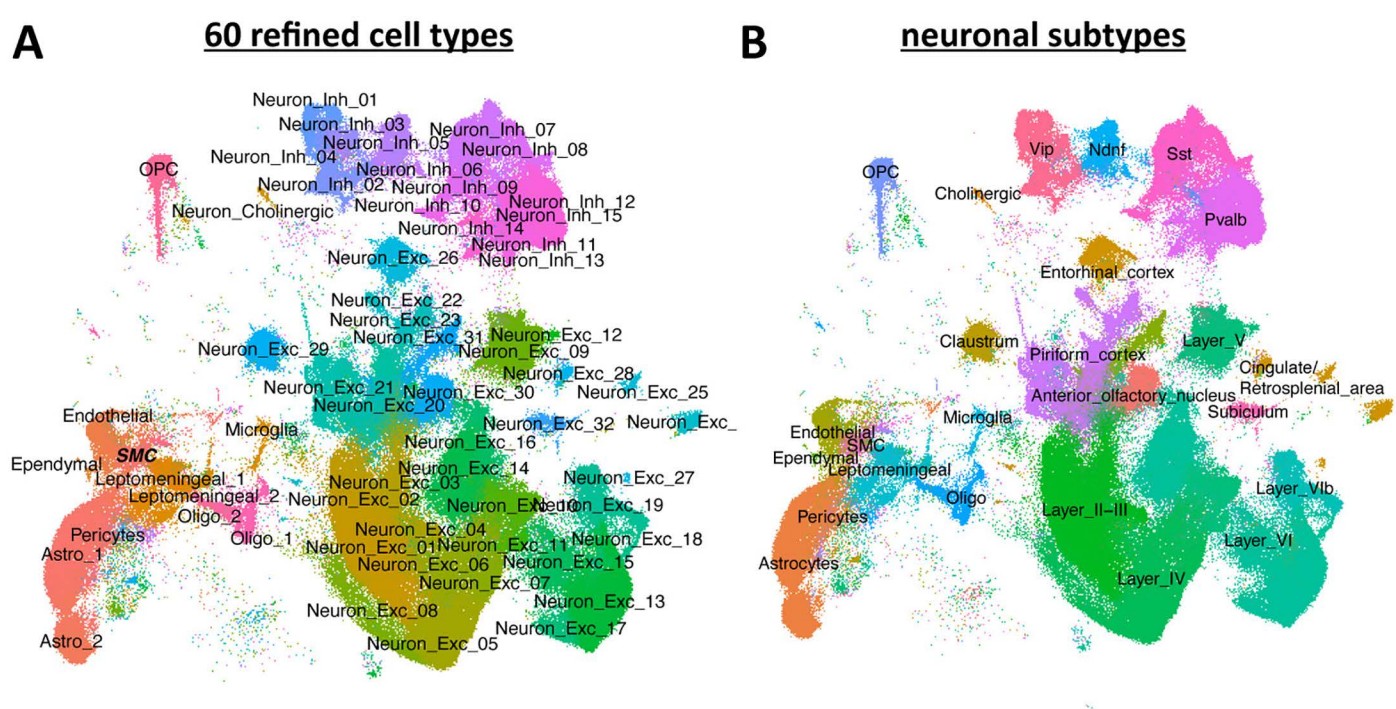

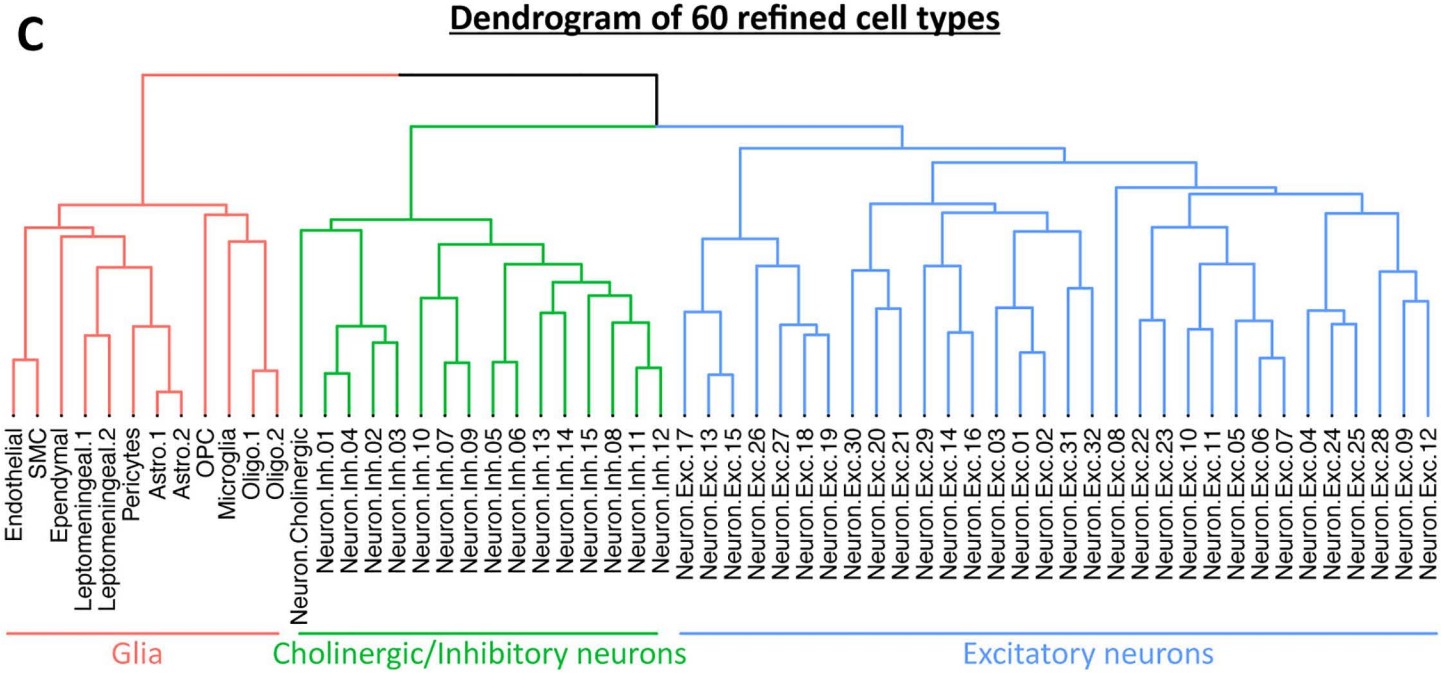

**Fig 2. Classification of 60 refined brain cell types in the CC mouse strains using snRNAseq. (A-B)** UMAP showing **(A)** 60 refined cell types and **(B)** neuronal subtypes. **(C)** Dendrogram showing similarity of 60 cell types based on average gene expression.

clusters with annotated cell types of published datasets (S2 Fig) [2, 4, 5]. We further classified excitatory and inhibitory neurons using layer-specific markers and neuropeptides, respectively (Fig 2B). Cell clusters that expressed markers indicative of non-cortical tissue origin were removed from downstream analysis, including one entire sample that had a high percentage of non-cortex tissue clusters (CC010 #3; S1H Fig). In total, we analyzed 287,047 nuclei from 92 animals that included over 1.5 billion UMIs (S1 Table).

### Analysis of cell type composition across the CC strains

We next computed the average proportion of each of the 12 principal cell types across all strains. Subsequently, we compared the proportion of each cell type observed in each CC strain to the corresponding average proportion across all strains (Fig 3A). Based on a one-way ANOVA, pericytes and endothelial cells were the only principal cell types showing a significant strain x cell proportion relationship (Table 1 and S1 File). We plotted the proportions of pericytes and endothelial cells across the CC strains and performed Tukey's HSD posthoc tests to determine which CC strains had statistically significant differences in proportion in these cell types (Fig 3B and 3C). The proportion of pericytes in CC025 was significantly increased compared to all other CC strains except for CC009. CC009 showed a significant increase in pericyte proportions compared to CC031, CC045 and CC046. For endothelial cells, CC025 showed statistically significant proportions compared to CC046 and CC084. We also evaluated changes in proportions of the 60 refined cell types across the CC strains (S3 Fig and S1 File) and found that 25 neuronal clusters, two leptomeningeal cell types, endothelial cells and pericytes showed statistically significant differences across the CC strains based on one way ANOVA (S2 Table). CC046 exhibited a reduced proportion of pericytes compared to CC009 and CC025, as well as a lower proportion of endothelial cells compared to CC025. Notably, CC046 showed an increased proportion of two excitatory neuronal clusters (Neuron_Exc_02 and Neuron_Exc_10) relative to other strains (S4 Fig). These two neuronal clusters had the lowest p-values in the one-way ANOVA test of refined cell types by strains, indicating the strongest strain-specific variation.

To examine potential transcriptional consequences of strain-specific shifts in vascular cell abundance, we conducted differential gene expression (DEG) analysis on pericytes and endothelial cells. We compared strains with high versus low proportions of each cell type, grouping CC009 and CC025 as high-pericyte/high-endothelial and CC045 and CC046 as low-pericyte/low-endothelial strains. Given the limited representation of these cell types in our dataset—pericytes comprising 0.165% and endothelial cells 1.62% of all cells—the number of cells available per strain for DEG analysis was constrained.

Nonetheless, DEG analysis using a stringent cutoff ($|log_2 FC| > 1.5$, adjusted p-value $< 0.05$) identified one differentially expressed gene in pericytes (*Hexb*) and 30 in endothelial cells (Fig 3D). Among these, several endothelial DEGs were enriched in the low-endothelial group (CC045 and CC046). *Hexb* emerged as a particularly interesting candidate; prior work has shown that endothelial-specific expression of *Hexb* can rescue neurological phenotypes in Sandhoff disease models [18], highlighting its key role in neurovascular function. Its upregulation in the low-endothelial strains may reflect a compensatory mechanism in response to compromised endothelial integrity.

We also observed elevated expression of *Il31ra* in the low-endothelial group. Given its established role in inflammatory signaling [19], this may point to enhanced immune–vascular interactions or impaired blood–brain barrier function in these strains—factors that could influence neural homeostasis. Together, these findings suggest that strain-specific deficits in vascular cell populations may trigger transcriptional adaptations with potential implications for brain function.

We next validated differences in pericyte and endothelial cell numbers by immunostaining brain sections from four CC strains that showed statistical differences in pericyte or endothelial cell composition, using markers of pericytes (CD13), vascular endothelial cells (PECAM-1) and nuclei (DAPI) (Fig 4A). We then quantified the number of pericytes (CD13 and DAPI double-positive puncta; Fig 4B), the number of endothelial cells (PECAM-1 and DAPI double-positive puncta; Fig 4C), the number of DAPI-positive cells, the CD13 positive area (a measure of pericyte area), and the PECAM-1 positive area (a measure of endothelial cell area; Fig 5E) in cerebral cortex sections from each animal using CellProfiler [20] (S5

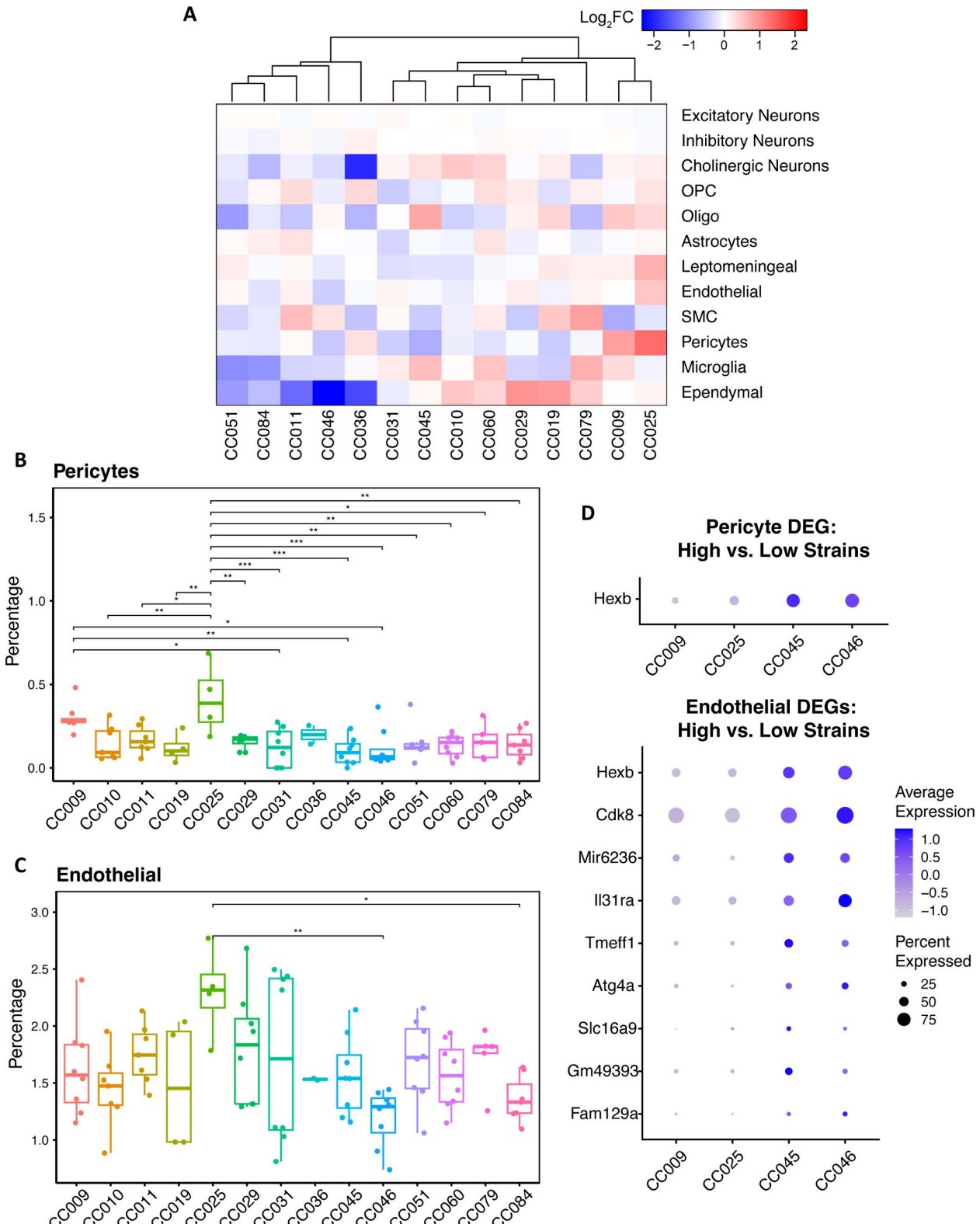

**Fig 3. Differences in principal cell type composition across the CC strains. (A)** Heatmap shows log2FC of each cell type for each strain, calculated by comparing to the average proportion of the corresponding cell type across all strains. **(B-C)** Boxplot showing cell composition of **(B)** pericytes and **(C)**

endothelial cells in each strain. Each dot represents each sample in a given strain. Statistical significance was calculated by Tukey's HSD posthoc tests. * p < 0.05; ** p < 0.01; *** p < 0.001. **(D)** Dotplot showing DEG(s) of high-pericyte/high-endothelial vs low-pericyte/low-endothelial strains in pericyte (top) or endothelial cell cluster (bottom).

**Table 1. One-way ANOVA result of principal cell type composition by strains.**

| Cluster | Effect | DFn | DFd | F | p | p.signif |
|---|---|---|---|---|---|---|
| Pericytes | Genotype | 13 | 78 | 3.807 | 9.98E-05 | *** |
| Endothelial | Genotype | 13 | 78 | 2.196 | 0.0172 | * |
| Leptomeningeal | Genotype | 13 | 78 | 1.753 | 0.0663 | |
| SMC | Genotype | 13 | 78 | 1.708 | 0.0756 | |
| Ependymal | Genotype | 13 | 78 | 1.449 | 0.157 | |
| Cholinergic Neurons | Genotype | 13 | 78 | 1.11 | 0.363 | |
| Inhibitory Neurons | Genotype | 13 | 78 | 0.962 | 0.495 | |
| Microglia | Genotype | 13 | 78 | 0.865 | 0.592 | |
| OPC | Genotype | 13 | 78 | 0.833 | 0.625 | |
| Excitatory Neurons | Genotype | 13 | 78 | 0.681 | 0.775 | |
| Oligo | Genotype | 13 | 78 | 0.645 | 0.808 | |
| Astrocytes | Genotype | 13 | 78 | 0.581 | 0.863 | |

Fig). Immunostaining results confirmed that pericyte counts were increased in CC009 compared to CC046 (Fig 5A). When pericyte counts were normalized to DAPI counts in the field of view, CC009 had increased DAPI-normalized pericyte proportions compared to CC045 and CC046, while CC025 had increased DAPI-normalized pericyte proportions compared to CC046 (Fig 5B). Endothelial cells also showed increased counts in CC009 and CC025 compared to CC045 and CC046 (Fig 5C and 5D). These histological studies further indicate that CC025 had significantly increased endothelial cell proportion compared to CC046, consistent with our snRNAseq cell proportion analysis.

As pericyte reduction might lead to pericyte stretching to cover more vasculature, we also quantified CD13 area over PECAM-1 area. We did not observe statistically significant changes across the strains for the quantification of CD13 occupied area over PECAM-1 occupied area (ANOVA p = 0.1198), suggesting that pericyte reduction did not lead to pericyte stretching in CC045 and CC046 strains (Fig 5F). We also quantified the ratio between pericytes and endothelial cells and observed that the ratios were not significantly different across the strains (Fig 5G).

## Discussion

Inbred C57BL/6 mice are commonly used to study brain development, behavior, and disease processes. This singular focus on one inbred strain can lead investigators to draw conclusions that may not generalize to other mouse strains or to other species, such as humans. Indeed, several studies have now shown that mouse genetic background can profoundly affect phenotypic variation and disease sensitivity. As one example, Tabbaa et al crossed a high-confidence autism gene mutation (*Chd8*) onto CC mice of different backgrounds and found that phenotypes in these strains varied substantially when compared to *Chd8* mutant mice on the C57BL/6 background [9]. In some strains, phenotypes were absent, weaker, or stronger relative to the C57BL/6 strain. Similarly, Schoenrock et al. demonstrated extensive behavioral variability in cocaine locomotor sensitivity and behavioral sensitization among CC strains and their founders.[21]. In addition, CC mice have also been used to reveal dramatic strain-dependent differences in seizure susceptibility and post-traumatic epilepsy following brain injury, as shown in two complementary studies by Gu and colleagues [11, 22]. These studies collectively

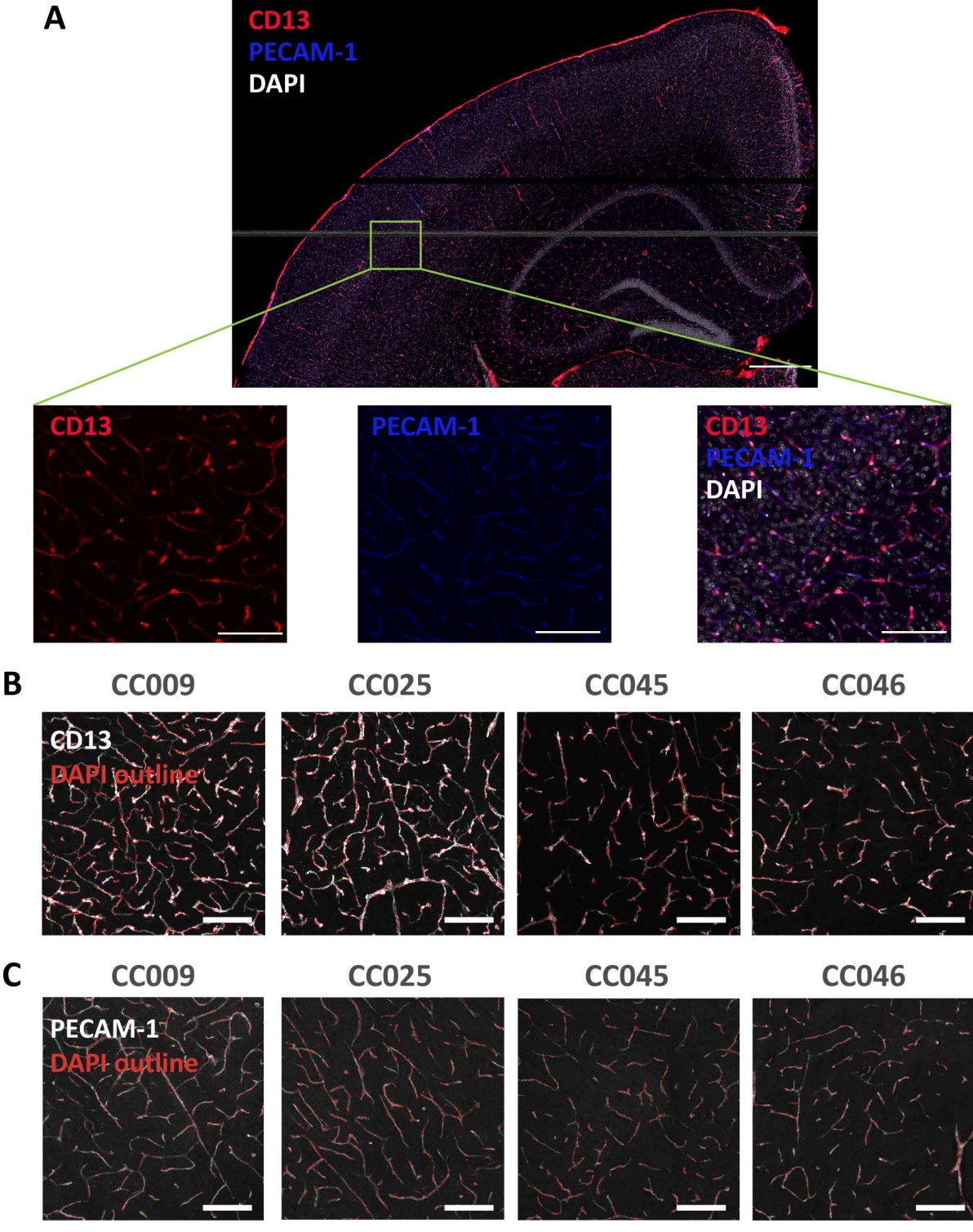

**Fig 4. Histological Visualization of Strain-Dependent Differences in Pericyte and Endothelial Cell Abundance in CC Mouse Cortex. (A)** Representative image of a CC mouse brain section stained with CD13, PECAM-1, and DAPI (scale bar = 500 μm). Several (n = 3-4) regions of interest (ROI) per section were selected in somatosensory motor area of cortex (zoomed image on the bottom; scale bar = 100 μm) for quantification. **(B-C)** Representative images highlighting **(B)** pericytes, defined by colocalization of CD13 and DAPI, and **(C)** endothelial cells, defined by colocalization of PECAM-1 and DAPI, across indicated strains. Quantification was based on 3 animals per strain, 2-3 sections per animal, and 2-4 ROIs per section.

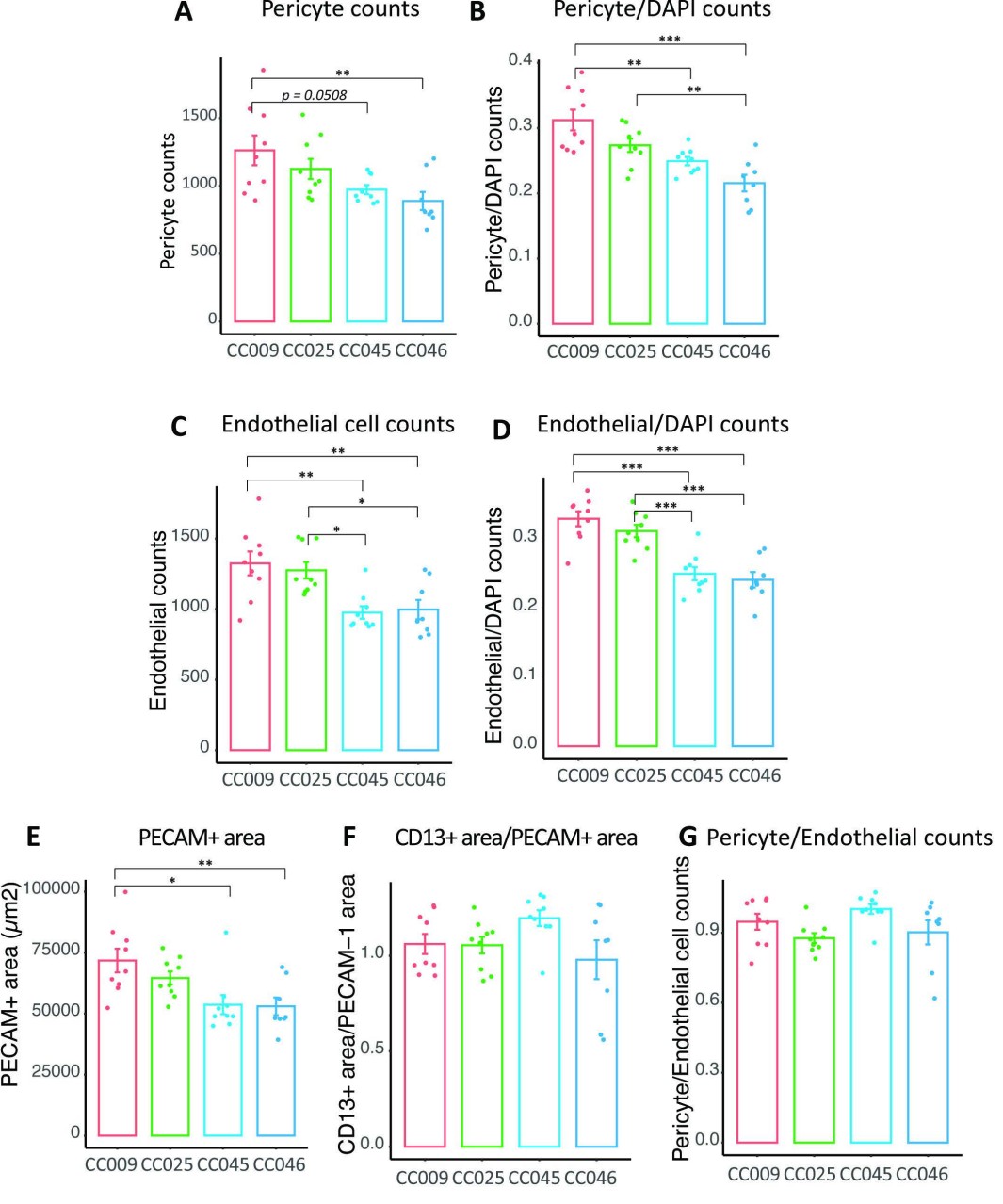

**Fig 5. Quantitative Analysis of Pericyte and Endothelial Cell Abundance from Histological Staining Across CC Strains. (A)** Quantification of pericyte counts (CD13 and DAPI double-positive areas) by strain. One-way ANOVA p-value: 0.0091. **(B)** Quantification of pericyte counts divided by DAPI counts from the indicated CC strains. One-way ANOVA p-value: 2.84e-05 **(C)** Quantification of endothelial cell counts (PECAM-1 and DAPI double-positive areas) by strain. One-way ANOVA p-value: 5.28e-04. **(D)** Quantification of endothelial cell counts divided by DAPI counts from the indicated CC strains. One-way ANOVA p-value: 3.68e-07. **(E)** Quantification of area occupied by PECAM-1 signal in each ROI image. One-way ANOVA p-value: 3.72e-03. **(F)** Quantification of area occupied by CD13 divided by area occupied by PECAM-1 signal in each ROI image. One-way ANOVA p-value: 0.11981. **(G)** Quantification of pericyte counts divided by endothelial cell counts. One-way ANOVA p-value: 0.0575. * p < 0.05; ** p < 0.01; *** p < 0.001.

demonstrate that genetic background can profoundly shape the manifestation and severity of diverse neurological phenotypes.

Given how genetic background can influence behavior, and how behavior is influenced by the brain, which is made up of different specialized cell types, we hypothesized that genetic background might also impact the cellular composition of the brain. With a focus on the cerebral cortex, we found that pericyte and endothelial cell proportions varied across genetically diverse mouse strains. Our snRNAseq data showed that CC009 mice had a statistically significant difference in pericyte cell composition compared to CC045 and CC046 mice, which we validated histologically. Likewise, our histological studies showed that pericyte and endothelial cell counts in CC025 mice were significantly increased compared to CC046 mice, aligning with our snRNAseq findings. Minor discrepancies between histological findings and snRNAseq data may stem from the smaller number of CC025 replicates (n = 4) used in the snRNAseq experiments when compared to the other strains (n = ~8). Additionally, histological analyses may provide greater sensitivity in detecting cell type differences across strains.

Pericytes envelop blood vessels and play a crucial role in maintaining vascular stability, promote angiogenesis, and contribute to the integrity of the BBB [23–26]. The identification of variations in pericyte proportions across diverse genetic backgrounds underscores the potential functional differences in pericyte-mediated and neurovascular processes.

Our findings align with previous research showing that genetic background influences vascular structure and remodeling. Wang et al [27] found that there was wide variation in the number and diameter of cerebral collateral vessels among some inbred mouse strains, showcasing the genetic impact on vasculature formation and remodeling. Additionally, another study reported differences in vessel density and pericyte coverage in the cortex among genetically diverse mouse strains [28]. Our unbiased study examining all brain cell types across a much larger number of strains, extends these observations, and shows that mouse genetic background has a measurable impact on neurovascular cells.

Strain variation in pericyte and endothelial cell number has several biological implications. Differences in the composition of these cell types could influence BBB penetration, impacting the permeability of small molecules and viruses. This is illustrated by studies examining the efficiency of AAV-PHP-B virus transduction to the central nervous system (CNS). The variability in Ly6a gene expression, modulated by the distinct genetic backgrounds of various inbred mouse strains, emerged as a contributing factor to these differences, underscoring the potential impact of genetic influences on BBB permeability [29]. In our snRNA-seq data, Ly6a expression was specifically enriched in endothelial cells compared to other cell types. Also, strain background has been shown to affect brain vascularization, collateral abundance, and sensitivity to brain ischemic injury and stroke [10, 30–32].

Beyond these considerations, our findings contribute to a broader understanding of neurovascular dysfunction in disease. Recent work highlighted neurovascular unit (NVU) dysfunction as a key pathological feature of Alzheimer's disease (AD), implicating pericytes in BBB disruption and neurodegeneration [33]. This study identified two distinct subtypes of human pericytes, T-pericytes and M-pericytes, characterized by their roles in solute transport and extracellular matrix organization, respectively. In our analysis, pericytes expressed several T-pericyte markers (*Slc6a1*, *Slc1a3*), suggesting that pericytes in our snRNA-seq mouse brain dataset more closely resemble human T-pericytes than M-pericytes (S6 Fig), consistent with findings in the literature [33]. Our study also reveals that the NVU is particularly sensitive to genetic background, raising the possibility that strain-dependent differences in pericyte and endothelial cell composition could contribute to genetic variability in neurovascular resilience or disease susceptibility. Furthermore, a recent review on neurovascular aging [34] underscores the link between NVU dysfunction and AD, emphasizing the role of pericytes in maintaining cerebrovascular integrity. Given that alterations in pericyte and endothelial cell numbers have been associated with vascular contributions to cognitive impairment and dementia, our findings suggest that genetic differences in NVU cell composition may be relevant to understanding susceptibility to AD and other neurovascular disorders.

Finally, our study highlights the broader impact of genetic background on neurodevelopment and disease. Recent work emphasized the association between exposure to neurotrophic viruses and an increased risk of neurodevelopmental disorders (NDD) [35]. The influence of genetic background on how these viruses affect individuals remains poorly

 

understood, underscoring the need for a deeper understanding of genetic influences on BBB function and permeability. Overall, our study reveals that genetic background can influence the proportion of certain brain cells. While our work focused on evaluating strain differences in male animals, future studies will be needed to evaluate the extent to which these strain differences are apparent in female mice of each strain, especially given recent studies showing how sex differences in one strain do not always replicate when examined in other strains [9].

## Materials and methods

All procedures used in this study were approved by the Institutional Animal Care and Use Committee at the University of North Carolina at Chapel Hill and carried out in accordance with relevant guidelines and regulations. The study is reported in accordance with ARRIVE guidelines.

### Mice

Mice were maintained three to five per cage on a 12 h:12 h light:dark cycle and given food and water *ad libitum*. Cerebral cortex samples for snRNAseq were collected from thirteen to fourteen weeks old male collaborative cross (CC) mice (2–8 animals/strain) after dissection for immediate single nuclei isolation. Sample size was decided based on the availability of each CC mouse strain. No statistical methods were used to predetermine sample size.

### Sample preparation for snRNAseq

To isolate nuclei from brain tissue, mice were deeply anesthetized with pentobarbital sodium (50 mg/kg, intraperitoneal injection). Anesthesia depth was confirmed by the absence of a response to toe-pinch. To minimize suffering, all animals were perfused transcardially with ice-cold phosphate-buffered saline (PBS) while under deep anesthesia. Perfusion served as the method of euthanasia. Following perfusion, the right cortical hemisphere was dissected and homogenized for nuclei isolation. Prior to tissue homogenization, the following buffers were made: NIM1, homogenization, and resuspension buffers. NIM1 buffer was made by mixing 1190 µl of distilled water, 250 µl of 1.5M sucrose, 37.5 µl of 1M KCl, 7.5 µl of 1M MgCl$_2$, 15 µl of 1M Tris Buffer, pH 8.0. Homogenization buffer was made by mixing 960 µl of NIM1 buffer, 1 µl of 1M DTT (Thermo Fisher Scientific), 10 µl of 40 U/µl RNase inhibitor (Enzymatics), 10 µl of 20 U/µl SUPERase-in RNase inhibitor (Thermo Fisher Scientific), 10 µl of 20 mg/ml BSA and 10 µl of 10% Triton X-100. Resuspension buffer was made by mixing 990 µl of PBS, 1.25 µl of 40 U/µl RNase inhibitor (Enzymatics), 2.5 µl of 20 U/µl SUPERase-in RNase inhibitor (Thermo Fisher Scientific), 10 µl of 20 mg/ml BSA and 5 µl of 10% Tween. Tenbroeck tissue homogenizer (Bellco Glass) was cleaned with distilled water, 100% ethanol and then distilled water and submerged in homogenization buffer. The tissue was homogenized using Tenbroeck tissue homogenizer and further homogenized by pipetting up and down 10–15 times using wide-bore pipette tips. Lysate (1.0 mL) was transferred into a DNA LoBind 1.5 mL tube (Eppendorf) and mixed. Homogenates was passively filtered with a 40 µm strainer (Falcon) into a 15 mL tube and then centrifuged at 500 x g for 5 minutes at 4°C. The supernatant was carefully aspirated and the nuclei were resuspended gently with wide-bore tip in 1.0 mL cold resuspension buffer. After centrifugation and aspiration of the supernatant, the nuclei suspension was passively filtered again with a 40 µm strainer into a 15 ml tube. The nuclei suspension was fixed on ice for 10 minutes by slowly adding 3.0 ml of 1.33% formaldehyde. After 10 minutes, 20 µl of 10% Tween was added to the suspension and centrifuged at 500 x g for 5 minutes at 4°C. The supernatant was carefully aspirated and the nuclei were resuspended in 1.8 mL cold resuspension buffer. After counting the nuclei, 200 µl of DMSO was added to the suspension and the isolated nuclei suspension was stored in −80°C until the library preparation step.

### snRNAseq

Libraries from nuclei were prepared using the Split Pool Ligation-based Transcriptome sequencing (SPLiT-seq) method [36] which utilizes a combinatorial barcoding technique to identify the cellular origin of RNA during snRNAseq analysis.

7,000 nuclei from each sample were used for library preparation. Cortical samples from a total of 92 CC mice were successfully processed. To minimize batch effects, the 92 CC mice were randomly divided into two groups, with each group containing a proportional representation of each CC strain. After randomization, the samples from each group were processed in two separate batches to ensure consistent processing conditions and minimize batch effects. All libraries were pooled and sequenced on an Illumina NovaSeq 6000-S4. Raw BCL files from the Illumina NovaSeq were demultiplexed into paired-end, gzip-compressed FASTQ files using bcl2fastq (Illumina) based on the index (4th barcode in the SPLiT-seq) sequences. Raw reads were filtered based on their quality using fastp [37] and processed via Alevin-fry [38], which utilizes Salmon [39] for parsing basic barcode and UMI and mapping of the reads to the constructed reference index. Barcode sequences (24 bp = three 8 bp barcode sequences) were extracted from R2 reads and R1 reads with unique molecular identifiers (UMIs) were pseudoaligned with structural constraints ("--sketch" option) to corresponding intronic or exonic reads in the splici (for spliced + intron) index assembled using a GENCODE M25 (GRCm38.p6) mouse reference genome. To filter cellular barcodes (CBs; 24 bp), the list of expected barcode sequences was given to Alevin-fry and only the barcodes that were within the hamming distance of 1 of the expected barcode sequences were retained.

The nucleus by gene matrix from Alevin-fry was imported to Seurat (v3.1.1) [40] in R. To filter out low-quality nuclei, nuclei expressing fewer than 1000 UMIs or 500 genes were removed. Nuclei expressing greater than 10% mitochondrial transcripts were also removed. Genes were removed if they were not expressed in at least 30 nuclei. The filtered Seurat object was divided by each animal after assigning each nucleus to the corresponding animal based on the first 8 bp barcode sequences, and normalized using SCTransform [41]. After normalization, all samples were merged and clustering and visualization of the merged dataset were performed using Uniform Manifold Approximation and Projection (UMAP), with the first 100 principal components (PCs) at a resolution of 2. The Seurat FindMarkers function was employed to discover genetic markers linked to distinct cellular subtypes. The classification of each cluster into specific cell types was achieved by utilizing the expression levels of well-established marker genes and by correlating our clusters to annotated cell clusters in other published datasets. To correlate our data to previously published scRNAseq datasets [2,4,5], average expression values were calculated for every gene from our dataset using Pearson residuals (Seuratobject@assays$SCT@scale.data). Raw data of other published datasets were downloaded from Gene Omnibus Expression (GEO) and processed using our pipeline to make the datasets comparable to each other. Genes were removed if they were not present in both datasets being compared. Pearson correlation coefficients were calculated between clusters from our study and clusters from published datasets. The resulting matrix of correlation coefficients were plotted as heatmaps.

## Histology

Brains were sectioned at 50 μm on a cryostat and treated as free-floating sections for the duration of the staining process. Sections not immediately used for staining were stored at −20°C in a cryoprotectant solution.

Three sections per brain (from approximately Bregma −1.82 mm) were selected for staining. Sections were rinsed in PBS, then switched to a Tris-buffered saline containing 2.7% sodium chloride with 0.3% Triton-X-100 (TBS/TX). After several rinses, a blocking solution containing 10% normal donkey serum (NDS; Genetex GTX73205) in TBS/TX was applied to the tissue for an hour. Sections were then incubated overnight, at room temperature, in a primary antibody cocktail prepared in NDS/TBS/TX and containing goat anti-CD13 (R&D Systems, AF2335; 1:200) and rat anti-CD31 (also known as PECAM-1; BD Bioscience 553370; 1:200). The next day, sections were rinsed again in TBS/TX, treated with NDS/TBS/TX for 30 min, and incubated for 5h at room temperature in a secondary antibody cocktail containing donkey anti-goat-Cy3 (Jackson ImmunoResearch 705-165-147, 1:500), donkey anti-rat-647 (Jackson ImmunoResearch 712-605-153, 1:500), and DAPI (Thermo Scientific 62248, 1:4,000) in NDS/TBS/TX. Sections were rinsed in TBS/TX, PBS, and then mounted onto Superfrost Plus slides (Fisher 12-550-15) and air-dried briefly before coverslipping with Fluoro-Gel Mounting Medium (Electron Microscopy Sciences 17985−10). Sections were imaged at 20X on Zeiss LSM 710.

## Quantitative PCR (qPCR)

Genomic DNA (gDNA) was extracted from mouse tail tissues. After adding 250 μl of digestion buffer and 5 μl of 10 mg/ml proteinase K, tissues were incubated in 55°C for at least 5 hours. The samples were placed in 94°C for 10 minutes to inactivate proteinase K. Then gDNA samples were precipitated using isopropanol and washed with 70% ethanol. After ethanol wash, gDNA was resuspended in 25 μl of molecular grade water. All gDNA samples were diluted to 25 ng/μl and 1 μl of gDNA was used for each 10 μl qPCR reaction. qPCR was performed with SsoAdvanced Universal SYBR Green Supermix (Bio-Rad Cat # 1725270) according to the manufacturer's protocol on a QuantStudio™ 5 Real-Time PCR System (Thermo Fisher Scientific). *Ube3a* gene primers were used to normalize the expression of sex chromosome genes, *Sry* and *Xist*. The sequences of the primers are: *Ube3a* F 5'-GAAGGCCATCACATATGCCAAAGG-3'; *Ube3a* R 5'-TGTCCCCAATGAAGAAGGGAGG-3'; *Sry* F 5'-TTGTCTAGAGAGCATGGAGGGCCATGTCAA-3'; *Sry* R 5'-CCACTCCTCTGTGACACTTTAGCCCTCCGA-3'; *Xist* F 5'-AAAGCCAAGGAGTGCTCGTA-3'; *Xist* R 5'-ACAAAGATTGGGCTGTCGAG-3'

## Image analysis/quantification

Confocal images were acquired with a Zeiss LSM 710 confocal microscope. Microscope settings were consistent across all strains. Images were imported into Fiji [42] and CellProfiler [43] for quantification. Using Fiji, Four regions of interest (ROI) within somatosensory cortex were selected for each section. Each ROI was split into three channels (DAPI, CD13, PECAM-1) and each channel image was saved as an individual bmp file (Fig 3A) and these images were used for downstream quantification analysis with CellProfiler. Using the "IdentifyPrimaryObjects" module in CellProfiler, we obtained outlines for DAPI, CD13 and PECAM-1 staining (Supplementary Fig 5). We then overlaid DAPI outline with either CD13 or PECAM-1 using the "MaskObjects" module and counted the number of DAPI+ and CD13+ profiles to obtain pericyte counts and the number of DAPI+ and PECAM-1+ to obtain endothelial cell counts. DAPI outlines were also used to quantify the number of nuclei per field of view. The sum of CD13 or PECAM-1 occupied area, obtained via the "MeasureImageAreaOccupied" module, was used to measure CD13 or PECAM-1 vessel length in each field of view. Using the "ExportToSpreadSheet" module, the quantified data was exported to excel spreadsheet for statistical analysis. Prior to quantification, z-stacks were flattened to a maximum intensity projection. Image quantification was performed by an investigator who was blinded to the CC strain information to minimize bias. After quantification, the blinding was removed for subsequent analysis.

## Statistical analysis

All results are presented as mean ± s.e.m. The differences between multiple groups were tested using ANOVA and statistical significance between two groups was calculated by Tukey's HSD posthoc tests. Results with $p < 0.05$ were considered to be statistically significant.

## Supporting information

**S1 Table. snRNAseq dataset summary.**
(XLSX)

**S2 Table. One way ANOVA result of refined cell types by strains.**
(XLSX)

**S1 File. Principal or refined cell type proportion by genotype or by animal.**
(XLSX)

**S1 Fig. Quality control measures for the snRNAseq data.** (A) Violin plot showing number of UMIs per nucleus across each CC strain. (B) Violin plot showing number of genes per nucleus across each CC strain. (C) Violin plot showing number of UMIs per nucleus across each principal cell type. (D) Violin plot showing number of genes per nucleus across each

principal cell type. (E) Scatter plot showing normalized counts for *Xist* (female-specific gene on X-chromosome) and four Y-chromosome genes (*Uty, Kdm5d, Ddx3y, Eif23y*) for each sample. Normalized counts are calculated by dividing the total counts of the genes of interest by the total counts of all expressed genes and multiplying by 10,000. (F) 2D density scatter plot showing normalized counts for *Xist* and the four Y-chromosome genes (*Uty, Kdm5d, Ddx3y, Eif23y*) for each nucleus for typical samples (representative sample CC019 #2 is shown) and an outlier sample (CC084 #3; presumed XXY male). (G) Quantitative PCR of *Sry* and *Xist* genes in two female, two male samples and CC083 #3 outlier sample. (H) Percentage of non-cortex tissue for each sample. One outlier sample CC010 #3 with a high percentage of non-cortex tissue is marked with an arrow.
(TIF)

**S2 Fig. Correlation between the current snRNAseq and published scRNAseq datasets.** Heatmap showing Pearson correlation between the average normalized gene expression from our brain cell types and those from Tasic et al. 2016, Zeisel et al. 2015 and Zeisel et al. 2018.
(TIF)

**S3 Fig. Composition of 60 refined cell types across the CC strains.** (A) Heatmap showing 60 refined cell type composition differences across the CC strains. Log2FC of each cell type for each strain was calculated by comparing to the average proportion of the corresponding cell type across all strains.
(TIF)

**S4 Fig. Two refined cell type composition with statistically significant differences across the CC strains.** (A-B) Boxplot showing cell composition of (A) Neuron_Exc_02 and (B) Neuron_Exc_10 in each strain. Each dot represents each sample in a given strain. Statistical significance was calculated by Tukey's HSD posthoc tests. * $p < 0.05$; ** $p < 0.01$; *** $p < 0.001$.
(TIF)

**S5 Fig. Approach used to quantify immunostained CC brain sections.** (A) Representative image of a CC brain cortex section. Green rectangles mark the regions of interest used to quantify areas positive for DAPI, CD13 and PECAM-1 staining. (B) Left panel shows DAPI signal from the immunostained brain section. Right panel shows DAPI positive area (purple) recognized by CellProfiler. (C) Left panel shows CD13 signal from the immunostained brain section. Right panel shows CD13 positive area (green) recognized by CellProfiler. (D) Left panel shows PECAM-1 signal from the immunostained brain section. Right panel shows PECAM-1 positive area (green) recognized by CellProfiler.
(TIF)

**S6 Fig. Expression of T- or M-pericyte markers within the pericyte cluster.** Dot Plot showing the expression of -pericyte markers (*Slc6a1, Slc1a3, Slc20a2, Slc12a7, Slc6a12, Slc6a13*) and M-pericyte markers (*Col4a1, Col4a2, Col4a3, Col4a4, Lama4, Adamts1*) within the identified pericyte cluster.
(TIF)

## Acknowledgments

We thank Zylka lab members for technical assistance.

## Author contributions

**Conceptualization:** Mark J. Zylka, Jieun Park.

**Data curation:** Jieun Park, Bonnie Taylor-Blake, James L. Krantz.

**Formal analysis:** Jieun Park.

**Funding acquisition:** Mark J. Zylka.

**Investigation:** Bonnie Taylor-Blake.

**Methodology:** Jieun Park, James L. Krantz.

**Project administration:** Jieun Park.

**Supervision:** Mark J. Zylka.

**Validation:** Jieun Park.

**Writing – original draft:** Jieun Park.

**Writing – review & editing:** Mark J. Zylka, Jieun Park, Bonnie Taylor-Blake, James L. Krantz.

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
