## [Decision Letter · Decision Letter 0]

14 May 2025

PONE-D-25-19046Single nucleus RNA sequencing of the cerebral cortex from genetically diverse inbred mouse strains reveals differences in pericyte and endothelial cell compositionPLOS ONE

Dear Dr. Zylka,

Thank you for submitting your manuscript to PLOS ONE. After careful consideration, we feel that it has merit but does not fully meet PLOS ONE’s publication criteria as it currently stands. Therefore, we invite you to submit a revised version of the manuscript that addresses the points raised during the review process. If you think you can address all the reviewers' points, you are welcome to submit a revised version. Importantly, your revised manuscript will be sent out for re-review. Please note that if changes do not meet editorial expectations, your manuscript will be rejected.

Please submit your revised manuscript in Jun 28 2025 11:59PM. If you will need more time than this to complete your revisions, please reply to this message or contact the journal office at plosone@plos.org . Please include the following items when submitting your revised manuscript:

We look forward to receiving your revised manuscript.

Kind regards,

Anju Vasudevan, PhD

Academic Editor

PLOS ONE

Journal Requirements:

2. To comply with PLOS ONE submissions requirements, in your Methods section, please provide additional information regarding the experiments involving animals and ensure you have included details on (1) methods of sacrifice, and (2) efforts to alleviate suffering.

“J.P. is supported by grants from the National Institute of Child Health and Human Development (NICHD; T32HD040127). Microscopy was performed at the UNC Neuroscience Microscopy Core (RRID:SCR_019060), supported, in part, by funding from the NIH-NINDS (P30 NS045892) and the NIH-NICHD (P50 HD103573). M.J.Z. is supported by grants from the NINDS, NIMH, and NIEHS (R01NS109304, R01MH120229, R35ES028366).”

“We thank Zylka lab members for technical assistance. J.P. is supported by grants from the National Institute of Child Health and Human Development (NICHD; T32HD040127). Microscopy was performed at the UNC Neuroscience Microscopy Core (RRID: SCR_019060), supported, in part, by funding from the NIH-NINDS (P30 NS045892) and the NIH-NICHD (P50 HD103573). M.J.Z. is supported by grants from the NINDS, NIMH, and NIEHS (R01NS109304, R01MH120229, R35ES028366).”

“J.P. is supported by grants from the National Institute of Child Health and Human Development (NICHD; T32HD040127). Microscopy was performed at the UNC Neuroscience Microscopy Core (RRID:SCR_019060), supported, in part, by funding from the NIH-NINDS (P30 NS045892) and the NIH-NICHD (P50 HD103573). M.J.Z. is supported by grants from the NINDS, NIMH, and NIEHS (R01NS109304, R01MH120229, R35ES028366).”

5. In the online submission form, you indicated that “Additional data supporting the findings of this study are available from the corresponding author upon reasonable request.”

Reviewers' comments:

Reviewer's Responses to Questions

**Comments to the Author**

1. Is the manuscript technically sound, and do the data support the conclusions?

Reviewer #1: Partly

Reviewer #2: Yes

2. Has the statistical analysis been performed appropriately and rigorously? 

Reviewer #1: Yes

Reviewer #2: Yes

3. Have the authors made all data underlying the findings in their manuscript fully available?

Reviewer #1: Yes

Reviewer #2: Yes

4. Is the manuscript presented in an intelligible fashion and written in standard English?

Reviewer #1: Yes

Reviewer #2: Yes

5. Review Comments to the Author

**Reviewer #1:**  The authors utilized snRNA-seq of a variety of different inbred mouse strains to investigate cell population differences. This is an important line of research to investigate, especially in regards to the variance observed across numerous different mouse strains. However, the author's dataset is under-analyzed with no form of gene expression analysis outside of cell type identification. Also, it is unclear how the authors could differentiate between CD13 and PECAM-1 positive nuclei at the magnification provided, with no representative images showing the differences across different strains. Thirdly, the conclusions that the authors have drawn are presumptuous and without sufficient data to support. Overall, this manuscript requires further work to aid the author's interpretations before suitable for publication.

Please see specific comments below.

Introduction

- Authors should mention relevant examples of recorded phenotypic differences between inbred mouse strains

Results

- Fig.4: Authors should show representative images for 4 strains being compared in figure.

- From the images shown in Fig.4, it is unclear how authors could differentiate between pericyte and endothelial nuclei due to the overlap of CD13 and PECAM-1 staining. Clearer images should be provided in supplement to support their analysis.

- The authors should analyze gene expression changes across strains to identify potential pathways that have been dysregulated by differing population numbers.

- Supplemental Fig.4: Differences in neuronal subtype population sizes are identified but authors should confirm by FISH or IHC, as they have attempted to do with the pericytes and endothelial cells.

Discussion

- Overly speculative without authors providing substantial evidence from their results. For example, authors identify neurovascular issues are associated with Alzheimer’s disease using Yang et al., 2022 and that their dataset expressed pericyte markers more closely aligned with T-pericytes. Yet, the authors did not present gene expression data outside of cell-type identification.

**Reviewer #2: ** The authors surveyed the cortical cellular composition of 14 Collaborative Cross mice and further validated the differences in pericyte and endothelial cells using IHC. This information is critical to understand the potential cellular mechanism behind phenotype diversity found in this population of mice. Some outstanding questions remain to be addressed:

What is the selection criteria of the 14 strains of CC. Are there particular neural trait of interest that the authors are looking into within this population of mice?

Why the excitatory and inhibitory neurons being the most conserved cell population across strains? Is it because they comprise the large majority of the cells and tolerant to any small changes if there's any?

Lack of control/reference strain. The authors mentioned the commonly used laboratory inbred mice in the neuroscience studies like B6J and 129. So it is important to know where do reference strains like B6J and 129 fall in the cell composition spectrum of principal/neuronal subtypes among the phenotyped CC? Can they take advantage of existing single nucleus seq data of B6J and compare to other CC strains?

Have any neuronal/behavioral phenotypes that have been identified in any of the CC009, CC025, CC046, CC045 that can be explained by the differential cellular composition?

Please cite and discuss other brain/vascular related phenotypic diversity that have been published in CC or their founder strains that pertain to the scope of this study: PMID: 34321020, PMID: 36275853, and PMID: 38185315.

6. PLOS authors have the option to publish the peer review history of their article (what does this mean? ). If published, this will include your full peer review and any attached files.

**Do you want your identity to be public for this peer review?** For information about this choice, including consent withdrawal, please see our Privacy Policy .

Reviewer #1: No

Reviewer #2: No

---

## [Author Response · Author response to Decision Letter 1]

26 Jun 2025

Response to Reviewers

Journal Requirements:

- We ensured that our manuscript meets PLOS ONE’s style requirements.

2. To comply with PLOS ONE submissions requirements, in your Methods section, please provide additional information regarding the experiments involving animals and ensure you have included details on (1) methods of sacrifice, and (2) efforts to alleviate suffering.

- We have updated the Methods section to describe the procedures used to sacrifice animals and minimize suffering. Perfusion under deep anesthesia was used as the method of euthanasia, in accordance with institutional guidelines and approved protocols.

Updated original methods text: “To isolate nuclei from brain tissue, mice were deeply anesthetized with pentobarbital sodium (50 mg/kg, intraperitoneal injection). Anesthesia depth was confirmed by the absence of a response to toe-pinch. To minimize suffering, all animals were perfused transcardially with ice-cold phosphate-buffered saline (PBS) while under deep anesthesia. Perfusion served as the method of euthanasia. Following perfusion, the right cortical hemisphere was dissected and homogenized for nuclei isolation.”

“J.P. is supported by grants from the National Institute of Child Health and Human Development (NICHD; T32HD040127). Microscopy was performed at the UNC Neuroscience Microscopy Core (RRID:SCR_019060), supported, in part, by funding from the NIH-NINDS (P30 NS045892) and the NIH-NICHD (P50 HD103573). M.J.Z. is supported by grants from the NINDS, NIMH, and NIEHS (R01NS109304, R01MH120229, R35ES028366).”

- We have included the following statement in our cover letter to address the journal’s requirement regarding the role of funders:

“J.P. is supported by grants from the National Institute of Child Health and Human Development (NICHD; T32HD040127). Microscopy was performed at the UNC Neuroscience Microscopy Core (RRID:SCR_019060), supported, in part, by funding from the NIH-NINDS (P30 NS045892) and the NIH-NICHD (P50 HD103573). M.J.Z. is supported by grants from the NINDS, NIMH, and NIEHS (R01NS109304, R01MH120229, R35ES028366). The funders had no role in study design, data collection and analysis, decision to publish, or preparation of the manuscript.”

“We thank Zylka lab members for technical assistance. J.P. is supported by grants from the National Institute of Child Health and Human Development (NICHD; T32HD040127). Microscopy was performed at the UNC Neuroscience Microscopy Core (RRID: SCR_019060), supported, in part, by funding from the NIH-NINDS (P30 NS045892) and the NIH-NICHD (P50 HD103573). M.J.Z. is supported by grants from the NINDS, NIMH, and NIEHS (R01NS109304, R01MH120229, R35ES028366).”

“J.P. is supported by grants from the National Institute of Child Health and Human Development (NICHD; T32HD040127). Microscopy was performed at the UNC Neuroscience Microscopy Core (RRID:SCR_019060), supported, in part, by funding from the NIH-NINDS (P30 NS045892) and the NIH-NICHD (P50 HD103573). M.J.Z. is supported by grants from the NINDS, NIMH, and NIEHS (R01NS109304, R01MH120229, R35ES028366).”

- We have also addressed the funding placement concern. The revised Acknowledgments section, with funding details removed, reads:

“We thank Zylka lab members for technical assistance.”

All funding-related information has been retained solely in the Funding Statement section, and we confirm that the current Funding Statement contains all necessary funding acknowledgments. Therefore, no further changes to the Funding Statement are required.

5. In the online submission form, you indicated that “Additional data supporting the findings of this study are available from the corresponding author upon reasonable request.”

- We have included all data underlying the findings described in the manuscript. This includes the raw and processed sequencing data, associated metadata, and analysis scripts, all of which are provided as stated in the Data and Code Availability section. As such, there is no additional data requiring deposition, and we have removed the statement, “Additional data supporting the findings of this study are available from the corresponding author upon reasonable request,” from the Data and Code Availability section.

The revised Data and Code Availability section now reads as follows (and is updated in the revised manuscript):

“The raw and processed sequencing data generated in this study have been deposited in the Gene Expression Omnibus (GEO) database under accession number GSE263649. The metadata and scripts used for snRNAseq analysis are available in the GitHub page (https://github.com/jieunesther/CC_snRNAseq/).”

Reviewer #1: The authors utilized snRNA-seq of a variety of different inbred mouse strains to investigate cell population differences. This is an important line of research to investigate, especially in regards to the variance observed across numerous different mouse strains. However, the author's dataset is under-analyzed with no form of gene expression analysis outside of cell type identification. Also, it is unclear how the authors could differentiate between CD13 and PECAM-1 positive nuclei at the magnification provided, with no representative images showing the differences across different strains. Thirdly, the conclusions that the authors have drawn are presumptuous and without sufficient data to support. Overall, this manuscript requires further work to aid the author's interpretations before suitable for publication.

Please see specific comments below.

Introduction

- Authors should mention relevant examples of recorded phenotypic differences between inbred mouse strains

We added the examples of recorded phenotypic differences between inbred mouse strains in the introduction (bolded parts are added).

In humans, functional genomics data from 1,866 brain samples revealed that several brain disorders were associated with alterations in brain cell type proportions (8). Different inbred mouse strains likewise show profound phenotypic differences in behavior and susceptibility to disease (9–13). For instance, Gu et al identified CC strains with extreme seizure resistance and SUDEP (sudden unexpected death in epilepsy) susceptibility and genomic loci associated with seizure sensitivity using standardized seizure assays and quantitative trait loci (QTL) mapping (PMID: 32852103). Similarly, inbred strains such as C57BL/6N and 129S2/SvPasCrl display marked differences in hippocampal gamma oscillation power and sharp wave-ripple events compared to C57BL/6J (PMID: 37550182). Additionally, anatomical variation in cortical area maps between C57BL/6J and DBA/2J highlights the influence of genetic background on brain structure (PMID: 15774010). While these phenotypic differences could, in part, relate to differences in brain cell composition, the extent to which strain background influences the types and proportions of cell types in the mouse brain has not been studied.

Results

- Fig.4: Authors should show representative images for 4 strains being compared in figure.

We added representative images from the four strains to Fig. 4.

- From the images shown in Fig.4, it is unclear how authors could differentiate between pericyte and endothelial nuclei due to the overlap of CD13 and PECAM-1 staining. Clearer images should be provided in supplement to support their analysis.

- We have now included representative images for each of the four strains analyzed to support our quantification of pericytes and endothelial cells. To clarify how pericytes and endothelial nuclei were distinguished, we highlight that pericytes were identified based on the colocalization of CD13 and DAPI signals, while endothelial cells were identified by the overlap of PECAM-1 and DAPI. These representative images have been added to the main figure panel (Figure 4B and 4C as shown).

- The authors should analyze gene expression changes across strains to identify potential pathways that have been dysregulated by differing population numbers.

- We focused our differential gene expression (DEG) analysis on pericytes and endothelial cells by comparing strains with high versus low proportions of each cell type. This targeted approach was taken due to the large number of potential comparisons, which are beyond the scope of a study specifically focused on vascular cell types. However, we have made all relevant data publicly available, so that other researchers can perform additional comparisons between strains of interest to explore broader differences in cell type proportion or transcriptional profiles.

To investigate whether specific pathways are dysregulated in strains with varying pericyte or endothelial cell proportions, we grouped CC009 and CC025 as high-pericyte/high-endothelial strains, and CC045 and CC046 as low-pericyte/low-endothelial strains. Given the relatively small representation of these cell types in our dataset—pericytes comprising 0.165% and endothelial cells 1.62%—the number of cells available for differential gene expression (DEG) analysis per strain was limited.

Despite this limitation, we performed DEG analysis between the high and low groups using a stringent threshold (|log₂FC| > 1.5, adjusted p-value < 0.05). This yielded one DEG in pericytes (Hexb) and 30 DEGs in endothelial cells. When visualized, several endothelial cell DEGs were specifically enriched in the low-endothelial strains (CC045 and CC046). Among these, Hexb is of particular interest: previous work has shown that endothelial-specific expression of Hexb can rescue neurological phenotypes in Sandhoff disease mouse models (Dogbevia et al., 2019), underscoring a critical role for Hexb in neurovascular health. Its upregulation in low-endothelial strains may reflect a compensatory response to impaired endothelial function.

Additionally, we observed elevated expression of Il31ra in the low-endothelial group. Given its involvement in inflammatory signaling, this may suggest enhanced immune–vascular crosstalk or impaired barrier integrity in these strains—potential contributors to altered neural homeostasis. These findings point toward transcriptional adaptations that may arise in response to strain-specific deficits in vascular cell populations.

To highlight transcriptional features enriched in strains with lower pericyte/endothelial cell proportions, we included representative DEGs from the low group in the dot plot shown in Figure 3D.

- Supplemental Fig.4: Differences in neuronal subtype population sizes are identified but authors should confirm by FISH or IHC, as they have attempted to do with the pericytes and endothelial cells.

- This analysis is beyond the scope of the current study. We prioritized validating the two most prominent cell composition differences observed between strains—pericytes and endothelial cells—through orthogonal methods. While neuronal subtype differences were also detected, they were of smaller magnitude. Further investigation of these differences can be examined in future studies.

- Overly speculative without authors providing substantial evidence from their results. For example, authors identify neurovascular issues are associated with Alzheimer’s disease using Yang et al., 2022 and that their dataset expressed pericyte markers more closely aligned with T-pericytes. Yet, the authors did not present gene expression data outside of cell-type identification.

- As noted in the Discussion, “pericytes expressed several T-pericyte markers (Slc6a1, Slc1a3), suggesting that pericytes in our snRNA-seq mouse brain dataset more closely resemble human T-pericytes than M-pericytes, consistent with findings in the literature (Yang et al., 2022).” However, we did not include supporting gene expression data in the original submission. To address this concern, we have now added a supplementary figure that shows the expression of representative T-pericyte markers (e.g., Slc6a1, Slc1a3, Slc20a2, Slc12a7, Slc6a13) and M-pericyte markers (Col4a1, Col4a2, Col4a3, Col4a4, Lama4, Adamts1) in our snRNAseq pericyte cluster (Supplementary Fig 6), showing stronger expression of some T-pericyte markers than M-pericyte markers. This addition strengthens our interpretation and provides the necessary evidence for the statement.

Reviewer #2: The authors surveyed the cortical cellular composition of 14 Collaborative Cross mice and further validated the differences in pericyte and endothelial cells using IHC. This information is critical to understand the potential cellular mechanism behind phenotype diversity found in this population of mice. Some outstanding questions remain to be addressed:

What is the selection criteria of the 14 strains of CC. Are there particular neural trait of interest that the authors are looking into within this population of mice?

- As we noted in the introduction, most single cell sequencing studies made use of one mouse strain (C57Bl/6). The extent to which brain cell composition differs as a function of strain background has never been explored before. The CC mice are housed at UNC. Periodically, UNC issues a call for pilot proposals to make use of the CC mice for new studies. We thus selected these 14 CC strains because all were white-coated (relevant to a different study we sought to perform) and were available for distribution as part of this pilot program. We thus initiated this study to address the primary question of how strain variation impacts brain cell composition, and to determine if genetic variation acts to a greater extent on certain brain cell types. Behavioral studies with these strains are well-beyond our goal. Our goal for this study was to determine if genetic variation among CC strains can contribute to measurable differences in brain cell type proportions. We are now the first to show that cells of the neurovascular unit are particularly sensitive to strain background.

Why the excitatory and inhibitory neurons being the most conserved cell population across strains? Is it because they comprise the large majority of t

---

## [Decision Letter · Decision Letter 1]

8 Jul 2025

Single nucleus RNA sequencing of the cerebral cortex from genetically diverse inbred mouse strains reveals differences in pericyte and endothelial cell composition

PONE-D-25-19046R1

Dear Dr. Zylka,

We’re pleased to inform you that your manuscript has been judged scientifically suitable for publication and will be formally accepted for publication once it meets all outstanding technical requirements.

Kind regards,

Anju Vasudevan, Ph.D

Academic Editor

PLOS ONE

Reviewers' comments:

Reviewer's Responses to Questions

**Comments to the Author**

1. If the authors have adequately addressed your comments raised in a previous round of review and you feel that this manuscript is now acceptable for publication, you may indicate that here to bypass the “Comments to the Author” section, enter your conflict of interest statement in the “Confidential to Editor” section, and submit your "Accept" recommendation.

Reviewer #2: All comments have been addressed

2. Is the manuscript technically sound, and do the data support the conclusions?

Reviewer #2: Yes

3. Has the statistical analysis been performed appropriately and rigorously? 

Reviewer #2: Yes

4. Have the authors made all data underlying the findings in their manuscript fully available?

Reviewer #2: Yes

5. Is the manuscript presented in an intelligible fashion and written in standard English?

Reviewer #2: Yes

6. Review Comments to the Author

Reviewer #2: All concerns have been addressed. Comparison to reference C57BL6J or 129 is highly recommended, or at least discussed.

7. PLOS authors have the option to publish the peer review history of their article (what does this mean? ). If published, this will include your full peer review and any attached files.

**Do you want your identity to be public for this peer review?** For information about this choice, including consent withdrawal, please see our Privacy Policy .

Reviewer #2: No

---

## [Editor Report · Acceptance letter]

PONE-D-25-19046R1

PLOS ONE

Dear Dr. Zylka,

I'm pleased to inform you that your manuscript has been deemed suitable for publication in PLOS ONE. Congratulations! Your manuscript is now being handed over to our production team.

Kind regards,

on behalf of

Dr. Anju Vasudevan

Academic Editor

PLOS ONE